# Expression of Basal Compartment and Superficial Markers in Upper Tract Urothelial Carcinoma Associated with Balkan Endemic Nephropathy, a Worldwide Disease

**DOI:** 10.3390/biomedicines12010095

**Published:** 2024-01-01

**Authors:** Ljubinka Jankovic Velickovic, Ana Ristic Petrovic, Zana Dolicanin, Slavica Stojnev, Filip Velickovic, Dragoslav Basic

**Affiliations:** 1Center for Pathology, University Clinical Center Nis, 18000 Nis, Serbia; slavicastojnev@gmail.com; 2Department of Pathology, Faculty of Medicine, University of Nis, 18000 Nis, Serbia; 3Department of Biomedical Sciences, State University of Novi Pazar, 36300 Novi Pazar, Serbia; dolicanin_z@yahoo.com; 4Department of Nuclear Medicine, Faculty of Medicine, University of Nis, 18000 Nis, Serbia; velickovicfilip@yahoo.com; 5Department of Urology, Faculty of Medicine, University of Nis, 18000 Nis, Serbia; basicdr@gmail.com

**Keywords:** upper tract urothelial cancer, Balkan endemic nephropathy, morphology, CKs, CD44

## Abstract

The aim of this study was to determine the association of basal compartment and superficial markers, comprising CK5/6, CD44, CK20, and the pathological characteristics of upper tract urothelial carcinoma (UTUC) associated with Balkan endemic nephropathy (BEN). Comparing the expression of the investigated markers in 54 tumors from the BEN region and 73 control UTUC, no significant difference between them was detected. In regression analysis, CK20 expression was not determined with expression of CK5/6, CD44, and the phenotypic characteristics of BEN and control UTUC. Parameters with predictive influence on the expression of CD44 in BEN UTUC included growth pattern (*p* = 0.010), necrosis (*p* = 0.019); differentiation (*p* = 0.001), and lymphovascular invasion (*p* = 0.021) in control UTUC. Divergent squamous differentiation in BEN tumors (*p* = 0.026) and stage in control tumors (*p* = 0.049) had a predictive influence on the expression of CK5/6. This investigation detected a predictive influence of the phenotypic characteristics of UTUC on the expression of basal compartment and superficial markers, with a significant influence of necrosis in BEN tumors (*p* = 0.006) and differentiation in control UTUC (*p* = 0.036).

## 1. Introduction

Upper tract urothelial carcinoma (UTUC) constitutes only 5–6% of all epithelial tumors of the urinary tract, but in some regions of the world, the incidence of urothelial neoplasms of the renal pelvis and ureter is quite high [1,2,3]. Balkan endemic nephropathy (BEN) is a chronic, slowly progressive tubulointerstitial disease, and in the area of BEN, UTUC may occur alone or in combination with BEN [4].

UTUC is more invasive and worse differentiated than bladder cancer; thus, it requires as precise as possible an assessment of disease progression and tumor invasiveness for every individual case. Factors such as age, tumor grade, stage, sessile tumor growth, lymphovascular invasion (LVI), lymph node involvement, necrosis, and tumor location have been reported in the literature to be associated with the prognosis of patients with UTUC [1,2,3,4]. 

Some investigations have suggested that differentiation in the majority of urothelial carcinomas mirrors normal urothelial differentiation [5]. CK20 is a marker of cellular differentiation and is considered a useful and reliable marker of neoplastic change in urothelial cells. In normal urothelium, CK20 expression is confined to umbrella cells and occasional intermediate cells [6,7,8]. On the other hand, CD44 and CK5/6 are markers of basal compartment in the urothelium [9]. Chan et al. [10] have described a tumor-initiating cell subpopulation in primary human bladder cancer based on the expression of markers similar to those of normal bladder basal cells (lineage CD44 + CK5 + CK20−). 

CD44 is a transmembrane glycoprotein involved in all essential cellular processes, like survival, differentiation, proliferation, migration, angiogenesis, and cellular signaling, through the presentation of cytokines and growth factors to the corresponding receptors. CD44 expression is associated with bladder cancer aggressiveness and resistance to chemo and radio treatment, and the ratio in urothelial cancer tissue and urinary exfoliated cells showed a significant correlation in the same patients; therefore, it was proposed as a prognostic predictor [9,11,12].

The aims of this study were to determine the association of basal compartment and superficial markers (CK5/6, CK20, and CD44) with the pathological characteristics of UTUC in BEN and a control population and to estimate the predictive impact of these markers in UTUC.

## 2. Patients and Methods

### 2.1. Patient’s Population

We studied 127 patients with UTUC who had undergone nephroureterectomy with bladder cuff removal and extended lymphadenectomy. All cases of UTUC were diagnosed at the Center for Pathology, University Clinical Center Nis. The study included 93 pelvic and 34 ureteral urothelial. Patients were divided into two groups: 54 patients were from villages along the South Morava River basin, which are endemic settlements for BEN (BEN tumors); and 73 were residents of areas that are free of BEN (control subjects). 

### 2.2. Histologic Analysis

The histological sections were processed from tissue fixed in 10% formalin and stained with hematoxylin and eosin (H&E). Obtained H&E slides were used to estimate histological variant, divergent differentiation, growth pattern (papillary/solid), tumor grade (low/high grade), and the presence of necrosis and lymphovascular invasion (LVI), as well as to determine pathologic stage (pT) [13]. The authors compared low-stage non-muscle invasive tumors (pTa and pT1) and high-stage muscle invasive (pT2-pT4) tumors [3]. The tumor necrosis was based on macroscopic and microscopic examination of the tumor, and the cut-off was the presence of 10% macroscopic necrosis confirmed microscopically [14]. The squamous differentiation was defined as the presence of intercellular bridges or keratinization [15].

### 2.3. Immunohistochemical Scoring

Monoclonal antibodies against CK 20, CK 5/6, and CD44 (Dako, Glostrup, Denmark) at dilution 1:50, 1:50, and 1:50, respectively, with a standard En Vision system were used. Slides were reviewed independently by three researchers (LJV, ARP, SS), and areas with greater positivity were selected. Cytoplasmic (CK20, CK5/6) and membranous (CD44) expression was recorded for the investigated antibodies.

Based on personal observations and findings derived from the previously reported literature, immunohistochemical expression of CK20, CK5/6, CD44 in UTUC was defined as normal or altered as follows.

CK20 immunoreactivity was classified as normal (N) group (expression in superficial cells or absent staining) and altered (A) group (focal pattern or diffuse pattern, in which more than 10% of tumor cells were positive) [11]. CK5/6 was classified as normal (N) group (no staining or staining only in basal/parabasal cells) and altered (A) group (moderate to strong staining usually through the full thickness of the urothelium) [7]. The CD44 normal (N) group included strong expression on the plasma membranes of basal cells or the immunoreactivity of CD44 in basal, suprabasal, and intermediate cells, but not in the superficial cells (accentuated pattern). The altered (A) group included tumors which displayed a focal or total loss of basal CD44 expression or a focal loss of staining in an otherwise-accentuated pattern [11].

According to the investigated antigens, all tumors were classified into four groups based on normal (N) or altered (A) expression of CK5/6/CD44/CK20. Altered expression of all three markers was detected in only 4/127 (3.14%) UTUC (two in BEN and two in control UTUC). Therefore, groups 1, 2, and 3 contain tumors with either normal or altered CK5/6 expression. Group 1 comprised tumors with altered CK20 (*n* = 26/127 (20.5%)), group 2 included tumors with alteration of both CK20 and CD44 (*n* = 39/127 (30.7%)), group 3 contained tumors with altered expression of CD44 (*n* = 32/127 (25.2%)), and group 4 comprised tumors with normal expression of all three markers (*n* = 26/127 (20.47%)).

### 2.4. Statistical Analysis

For the purposes of analysis, pathological tumor stage (low vs. high), grade (low vs. high), growth pattern (papillary vs. solid), LVI (no vs. yes), necrosis (no vs. yes), squamous differentiation (no vs. yes), and clinical parameters, i.e., sex (M vs. F) and localization (pelvis vs. ureter), were evaluated as dichotomized variables. The χ^2^ (Fisher’s exact) test was used to estimate the expression of CK20, CK5/6, and CD44 in regard to pathological parameters (stage, grade, growth pattern, LVI, necrosis, squamous differentiation of tumors). Logistic regression analysis was used to detect the influence of every morphological characteristic, respectively, and each separately to the expression of CK5/6, CD44, and CK20. PLUM (Polytomous Universal Model) regression analysis was used to detect the predictive influence of the investigated pathological characteristics on the expression of basal compartment and superficial markers.

The result was considered statistically significant if *p* < 0.05. All analyses were performed with the Statistical Package for Social Sciences (version 24.0 statistical software (SPSS, Chicago, IL, USA).

## 3. Results

### 3.1. Clinical Features in UTUC

The age of the 127 patients with UTUC ranged from 22 to 85 years, with a mean age of 64.74 ± 8.31 years for tumors in BEN regions and 63.89 ± 10.7 years for control tumors. There were 25 male (46%) and 29 female (54%) patients in the BEN-associated UTUC group with ratio M:F = 1.2:1; while in the control group, there were 39 men (53%) and 34 women (47%) with ratio M:F = 1.1:1. With respect of localization, tumors were more frequent localized on the left side in both BEN and control UTUC, albeit without a statistical difference between these two groups (32/22 versus 46/27).

### 3.2. Immunohistochemical Evaluation of CKs and CD44 and the Association with Pathological Characteristics in BEN and Control UTUC 

The investigated markers—CK20, CK5/6, and CD44—were altered in 65 (51.2%), 14 (11%), and 71 (55.9%) UTUC, respectively (Figure 1).

Through investigation of the relationships between conventional pathological parameters and the altered immunohistochemical staining of CK20, CK5/6, and CD44 in UTUC, BEN tumors showed that altered expression of CK20 was significantly associated with grade (high 22/10 (68.8%) versus low 9/13 (40.9%), χ^2^ = 4.06, *p* < 0.05); and CD44 was significantly linked to tumor grade, stage, growth, and presence of necrosis (high-grade 24/8 (75%) versus low grade 7/15 (31.8%), χ^2^ = 9.76, *p* < 0.005; high-stage 22/9 (71%) versus low-stage 9/14 (39.1%), χ^2^ = 5.37, *p* < 0.05; solid growth 28/8 (77.8%) versus papillary 3/15 (16,7%), χ^2^ = 17.99, *p* < 0.00005; necrosis (yes 20/2 (90.9%) versus no 11/21 (34.4%), χ^2^ = 16.73, *p* < 0.00005)) (Table 1). 

Control tumors displayed a statistically significant association between altered expression of CK20 and LVI (LVI yes 15/8 (65.2%) versus LVI no 19/31 (38%), χ^2^ = 4.63, *p* < 0.05); CD44 was in statistically significant association with grade, stage, growth, and LVI (high-grade 34/14 (70.8%) versus low-grade 6/19 (24%), χ^2^ = 14.36, *p* < 0.0005; high-stage 32/18 (64%) versus low-stage 8/15 (34.8%), χ^2^ = 5.35, *p* < 0.05; solid growth 31/15 (67.4%) versus papillary 9/18 (33.3%), χ^2^ = 7.86, *p* < 0.005; LVI yes 19/4 (82.6%) versus LVI no 21/29 (42%), χ^2^ = 10.34, *p* < 0.005). A significant association was not detected between the phenotypic characteristics of BEN and control UTUC and altered expression of CK5/6 (Table 1).

Comparing the expression of CK20, CK5/6, and CD44 and group, a significant difference was not detected between BEN and control tumors. However, BEN tumors with necrosis had a significant difference in altered expression of CD44 compared to control tumors with the same morphological findings (20/2 (90.9%) versus 21/12 (63.6%), χ^2^ = 5.08, *p* < 0.05). 

### 3.3. Influence of Expression of Basal Compartment and Superficial Markers-CKs and CD44 on Pathological Characteristics of BEN and Control UTUC

BEN tumors contain 11/54 (20.37%) UTUC from group 1; 20/54 (37.03%) from group 2; 11/54 (20.37%) from group 3; and 12/54 (22.2%) from group 4 UTUC. Group 2 is significantly differentiated from group 4 in grade, stage, growth, and necrosis (high-grade, χ^2^ = 12.92, *p* < 0.0005; high-stage, χ^2^ = 5.91, *p* < 0.05; solid growth, χ^2^ = 16.67, *p* < 0.00005; and presence of necrosis, χ^2^ = 9.74, *p* < 0.005, respectively). Group 2 UTUC is very similar to that of group 1 and differentiated in terms of growth (χ^2^ = 4.94, *p* < 0.05), and no significant difference was detected between groups 2 and 3. Also, tumors from group 3 are significantly differentiated from UTUC group 4 in terms of grade, stage, growth, and necrosis (χ^2^ = 12.06, *p* < 0.001; χ^2^ = 5.01, *p* < 0.05; χ^2^ = 12.13, *p* < 0.0005; and χ^2^ = 15.43, *p* < 0.0001, respectively). Group 1 UTUC, with alteration of superficial marker CK20, has a significant difference in grade compared with group 4 (χ^2^ = 7.40, *p* < 0.001) (Table 2).

Control tumors account for 15/73 (20.54%) UTUC from group 1; 19/73 (26.02%) from group 2; 21/73 (28.76%) from group 3; and 14/73 (19.17%) from group 4 UTUC. In group 2 of control UTUC, compared to group 4, there was a significant difference in grade, stage, growth, and LVI was detected (χ^2^ = 21.19, *p* < 0.000005; χ^2^ = 4.45, *p* < 0.005; χ^2^ = 12.61, *p* < 0.0005; χ^2^ = 12.02, *p* < 0.0015, respectively). Group 2 is very similar to group 3 (difference only in LVI, χ^2^ = 6.19, *p* < 0.05), but the difference was evident between this group and group 1 in terms of grade, stage, and LVI (high-grade, χ^2^ = 7.75, *p* < 0.01; high-stage, χ^2^ = 3.93, *p* < 0.05, and presence of LVI, χ^2^ = 10.01, *p* < 0.005). UTUC with alteration of CD44 (group 3) differed significantly from group 1 in grade (high grade, (χ^2^ = 4.45, *p* < 0.05), and differed from group 4 in grade and growth (high grade, χ^2^ = 17.00, *p* < 0.00005; solid growth, χ^2^ = 8.17, *p* < 0.005, respectively) (Table 2).

In addition, the multistep logistic regression model, which included investigated basal and superficial markers (CK5/6, CD44, CK20), as well as group, localization, and pathological characteristics, showed that alteration of CK20 was not determined by expression of CD44 and CK5/6 and the phenotypic characteristics of UTUC and the groups (BEN and control UTUC). Differentiation and LVI had a prominent influence on the expression of CD44 in control UTUC (Wald = 10.464 p = 0.001; Wald = 5.316 *p* = 0.021). Growth pattern and necrosis had the prominent influence on the expression of CD44 in BEN UTUC (Wald = 6.654 *p* = 0.010; Wald = 5.460 *p* = 0.019). Squamous divergent differentiation in BEN UTUC had a notable influence on the expression of CK5/6 (Wald = 4.974 *p* = 0.026), and stage in the control UTUC determined the expression of CK5/6 (Wald = 3.890 *p* = 0.049) (Table 3).

PLUM regression analysis of expression CK5/6, CD44, and CK20 in control UTUC showed that differentiation had a predictive influence on expression of the basal compartment and superficial markers (Wald = 4.404, *p* = 0.036) and on necrosis in BEN tumors (Wald = 7.707, *p* = 0.006) (Table 4).

## 4. Discussion

In the urothelium, the transition from basal to terminally differentiated superficial cells is reflected in the different protein synthesis for each layer, which leads to the various morphological and antigenic characteristics [11,16]. CD44 is a cell adhesion molecule involved in tumor growth and biological behavior. CD44 realizes its functions through Fas inhibition, activating the mitogen-activated protein kinase (MAPK), and signaling and regulating the matrix metalloproteinases (MMPs) [17,18,19,20]. Recent data suggested that CD44 and p53 are important markers for the differential diagnosis of CIS from reactive/normal urothelium [11]. Inactivation of p53 results in overexpression of CD44, which may act as a tumor-promoting agent. However, the complexity of the problem stems from the poorly explained dualistic nature of CD44, which was found to be implicated in both tumor suppression and tumor promotion [21].

Our study showed that the dominant pattern of CD44 expression was altered, i.e., through the loss of CD44, in 55.9% of UTUC. The loss of CD44 expression is significantly connected with the morphological characteristics of aggressiveness in both control and BEN UTUC, which is reflected in high-grade, muscle-invasive disease, and the solid architecture pattern, as well as the presence of LVI in control tumors and necrosis in BEN UTUC.

Parameters with predictable influence on the expression of CD44 in BEN UTUC included architectural pattern and necrosis. Our previous comparative morphological study of BEN-associated tumors identified the sessile tumor architecture as a particular characteristic of these tumors [4], and this investigation detected that the solid growth of BEN tumors determined the loss of CD44 antigen. A predictable influence on the expression of CD44 in control UTUC was had by WHO grade and LVI. Similar findings related to the correlation of CD44 immunoreactivity and WHO grade, differentiation and LVI, have been reported by others [22,23].

CK5/6 is identified in squamous epithelium, the basal cells of the prostate, myoepithelial cells, and in different epithelial neoplasms [24]. In our study, parameters that had predictable influence on the expression of CK5/6 were squamous differentiation in BEN UTUC and stage of the control tumor, which has been shown in regression analysis [25]. Our previous morphological study of BEN tumors detected a higher frequency of divergent changes in BEN UTUC than in control tumors [4]. Additionally, urothelial lesions with squamous features showed higher CK5/6 expression. The CK 5/6 staining pattern varied between well-differentiated and poorly differentiated urothelial carcinoma. In low-grade papillary urothelial neoplasms, the CK 5/6-positive cells were observed at the basal cells; whereas in high-grade urothelial carcinoma, tumor cells were diffusely positive for CK 5/6 [24,26,27].

High expression of superficial marker CK20 was significantly associated with high grade in BEN tumors and the presence of LVI in control UTUC, but in regression analysis, the morphology of UTUC did not have a predictive influence on the expression of CK20, or the expression of CK5/6 and CD44.

BEN and control tumors have a similar presentation of basal compartment and superficial antigen. On the other hand, the absence of these antigens in BEN tumors is reflected in a significant presence of necrosis in regard to tumors with coexpression and heterogenous expression of the investigated antigens; moreover, the loss of coexpression in control tumors is associated with high grade in regard to other antigens profiles. In regression analysis, we detected that necrosis in BEN tumors and differentiation in control UTUC had a significant predictive influence on the change in the antigenic profile, from coexpression to the loss of coexpression, of basal compartment and superficial antigens.

This investigation showed that BEN and control tumors have a similar antigen presentation of basal compartment and superficial layer. Our findings indicate a predictive influence of the phenotypic characteristics of UTUC on the expression of basal compartment and superficial markers, with a significant influence of necrosis in BEN tumors and differentiation in control UTUC.

## Figures and Tables

**Figure 1 biomedicines-12-00095-f001:**
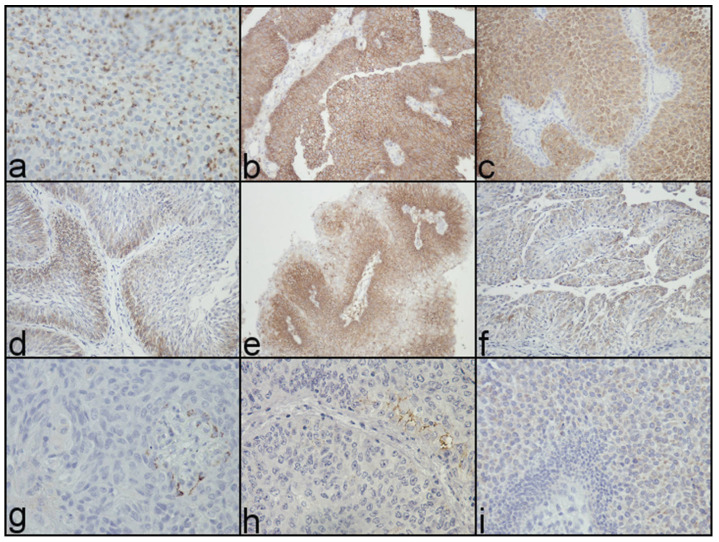
The representative altered and normal immunohistochemical expression of basal compartment and superficial markers (CK5/6 (**a**,**d**,**g**), CD44 (**b**,**e**,**h**), and CK20 (**c**,**f**,**i**)) in BEN-associated upper tract urothelial carcinoma with coexpression patterns (**a**,**b**,**c**) (original magnification: ×400).

**Table 1 biomedicines-12-00095-t001:** Association of CK20, CK5/6, and CD44 expression with pathological characteristics of BEN and control tumors.

UTUC	BEN N54	CK20	CK5/6	CD44	Control N73	CK20	CK5/6	CD44
Grade								
Low	22	9	1	7	25	8	3	6
High	32	22	4	24	48	25	6	34
*p*<		0.05	NS *	0.005		NS *	NS *	0.0005
Stage								
Low	23	11	1	9	23	8	6	8
High	31	20	4	22	50	26	3	32
*p*<		NS *	NS *	0.05		NS *	NS *	0.05
Growth								
Papillary	18	7	1	3	27	9	3	9
Solid	36	24	4	28	46	25	6	31
*p*<		NS *	NS *	0.00005		NS *	NS *	0.005
LVI								
no	42	23	4	28	50	19	7	21
yes	12	8	1	9	23	15	2	19
*p*<		NS *	NS *	NS *		0.05	NS *	0.005
Necrosis								
no	32	18	1	11	40	17	5	19
yes	22	13	4	20	33	17	4	21
*p*<		NS *	NS *	0.00005		NS *	NS *	NS *
Divergent dif.								
no	40	23	1	22	53	24	4	27
yes	14	8	4	9	20	10	5	13
*p*<		NS *	NS *	NS *		NS *	NS *	NS *

* NS: not significant.

**Table 2 biomedicines-12-00095-t002:** Expression of basal compartment and superficial markers in BEN and control UTUC.

Expression of CK5/6, CD44, and CK20												
UTUC	BEN N 54						Control N73					
	1:2	1:3	1:4	2:3	2:4	3:4	1:2	1:3	1:4	2:3	2:4	3:4
Grade *p*<	4/7 5/15 NS *	4/7 2/9 NS *	4/7 11/1 0.01	5/15 2/9 NS *	5/15 11/1 0.0005	2/9 11/1 0.001	7/8 1/18 0.001	7/8 3/18 0.05	7/8 12/2 0.05	1/18 3/18 NS *	1/18 12/2 0.000005	3/18 12/2 0.00005
Stage *p*<	5/6 6/14 NS *	5/6 3/8 NS *	5/6 9/3 NS *	6/14 3/8 NS *	6/14 9/3 0.05	3/8 9/3 0.05	6/9 2/17 0.05	6/9 6/15 NS *	6/9 6/8 NS *	2/17 6/15 NS *	2/17 6/8 0.05	6/15 6/8 NS *
Growth *p*<	5/6 2/18 0.05	5/6 1/10 NS *	5/6 10/2 NS *	2/18 1/10 NS *	2/18 10/2 0.00005	1/10 10/2 0.0005	6/9 3/16 NS *	6/9 6/15 NS *	6/9 11/3 0.05	3/16 6/15 NS *	3/16 11/3 0.0005	6/15 11/3 0.005
LVI *p*<	10/1 13/7 NS *	10/1 9/2 NS *	10/1 10/2 NS *	13/7 9/2 NS *	13/7 10/2 NS *	9/2 10/2 NS *	13/2 6/13 0.005	13/2 15/6 NS *	13/2 13/1 NS *	6/13 15/6 0.05	6/13 13/1 0.001	15/6 13/1 NS *
Necrosis *p*<	9/2 9/11 NS *	9/2 2/9 0.005	9/2 12/0 NS *	9/11 2/9 NS *	9/11 12/0 0.005	2/9 12/0 0.0001	8/7 9/10 NS *	8/7 10/11 NS *	8/7 10/4 NS *	9/10 10/11 NS *	9/10 10/4 NS *	10/11 10/4 NS *
Squamous differentiation *p*<	8/3 15/5 NS *	8/3 7/2 NS *	8/3 10/2 NS *	15/5 7/4 NS *	15/5 10/2 NS *	7/4 10/2 NS *	11/4 13/6 NS *	11/4 14/7 NS *	11/4 13/1 NS *	13/6 14/7 NS *	13/6 13/1 NS *	14/7 13/1 NS *

1. CK5/6 N and A, CD44 N, CK20 A; 2. CK5/6 N and A, CD44 A, CK20 A; 3. CK5/6 N and A, CD44 A, CK20 N; 4. CK5/6 N, CD44 N, CK20 N. * NS: not significant.

**Table 3 biomedicines-12-00095-t003:** Logistic regression analysis of basal compartment and superficial markers, CK5/6, CD44, and CK20, and morphological characteristics in UTUC.

Basal Compartment and Superficial Markers								
Dependent Variable	Variable	B	S.E.	Wald	Sig.	Exp(B)	95.0% C.I. for EXP(B)	Dependent Variable
CK20	CD44	−0.344	0.359	0.917	0.338	0.709	0.350	1.434
	CK5/6	0.384	0.574	0.447	0.504	1.468	0.476	4.522
Morphological characteristics: BEN UTUC								
CK20	All entered variables **				NS ***			
CD44	GROWTH	−3.364	1.304	6.654	0.010	0.035	0.003	0.446
	NECROSIS	−2.530	1.083	5.460	0.019	0.080	0.010	0.665
	All others				NS ***			
CK5/6	DIFF	−2.973	1.333	4.974	0.026	0.051	0.004	0.698
	All others	−3.364	1.304	6.654	0.010	0.035	0.003	0.446
Morphological characteristics: CONTROL UTUC								
CK20	All entered variables **				N.S.			
CD44	LG/HG	−3.795	1.173	10.464	0.001	0.022	0.002	0.224
	LVI	−1.879	0.815	5.316	0.021	0.153	0.031	0.754
	All others				NS ***			
CK5/6	STAGE	2.365	1.199	3.890	0.049	10.639	1.015	111.527
	All others *				NS ***			

* All entered variables: group, localization (pyelon/ureter), LG/HG, stage, growth, LVI, necrosis, divergent differentiation. ** All entered variables: P/U, LG/HG, stage, growth, LVI, necrosis, divergent differentiation. *** NS: not significant.

**Table 4 biomedicines-12-00095-t004:** PLUM regression analysis of expression of basal compartment and superficial markers, CK5/6, CD44, and CK20, and morphological characteristics in UTUC.

		Estimate	Std. Error	Wald	Sig.	95% Confidence Interval	
						Lower Bound	Upper Bound
BEN UTUC							
Threshold	[Group = 1–2]	−2.651	1.675	2.506	0.113	−5.934	0.631
	[Group = 2–3]	0.763	1.655	0.213	0.645	−2.481	4.007
Location	NECROSIS	2.278	0.821	7.707	0.006	0.670	3.886
	All others *				NS **		
CONTROL UTUC							
Threshold	[Group = 1–2]	−0.349	1.195	0.085	0.770	−2.690	1.993
	[Group = 2–3]	2.062	1.228	2.816	0.093	−0.346	4.469
Location	DIF	1.523	0.726	4.404	0.036	0.101	2.946
	All others *				NS **		

* All entered variables: P/U, dif., stage, growth (pap/sol), LVI (no/yes), necrosis (no/yes), divergent diff (no/yes). ** NS: not significant.

## Data Availability

Data are contained within the article.

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
