# Peer review of "Expression of Basal Compartment and Superficial Markers in Upper Tract Urothelial Carcinoma Associated with Balkan Endemic Nephropathy, a Worldwide Disease"

_biomedicines, 2024, doi:10.3390/biomedicines12010095_

Round 1
Reviewer 1 Report
Comments and Suggestions for Authors
1. This study demonstrated the association between basal compartment and superficial makers in UCC of upper tract with Balkan endemic nephropathy with phenotypic characteristics.
2. However, this manuscript presented with complicated and verbose writing in methods. The classification of groups in markers should be simplified and clarified.
3. The aims of this study were to investigate influence of basal compartment markers of UTUC in BEN and control population.
How about these markers expression in normal urothelium near UCC in BEN ? It is better to use normal urothelium of BEN as control ?
4. In patients and method: Does it justify to define patients from endemic settlements, the villages along the South Morava River basin as BEN tumors? Residents of rural andcity areas treat as free of BEN ?
5. In histology analysis : How to define necrosis ?
6. The term TCC is not used currently instead of UCC .
7. The classification and grouping are really difficult to understand and be rational.
8. Tables are also poorly manifest . Table one and two with different format. Table 2 dif. st . not explain in footnote .
9. Table 3&4 must be improved in format.
Author Response
Dear Reviewer 1,
Thank you very much for all the comments and suggestions. All the revisions made will contribute to a better quality of the manuscript. We have done Major revisions and we have changed everything you asked for.
- This study demonstrated the association between basal compartment and superficial makers in UCC of upper tract with Balkan endemic nephropathy with phenotypic characteristics.
Response: Patients were divided in two groups: 54 patients were from endemic settlements, the villages along the South Morava River basin without BEN (BEN tumors) and 73 control subjects, residents of settlements rural and city areas free of BEN which are closely linked to endemic areas.
We demonstrated the association between basal compartment and superficial makers in UCC of upper tract of patients from BEN settlements with UCC of upper tract from the residents from areas that are closely linked to BEN, but free of BEN.
- However, this manuscript presented with complicated and verbose writing in methods. The classification of groups in markers should be simplified and clarified.
Response: Method have been rewritten.
- The aims of this study were to investigate influence of basal compartment markers of UTUC in BEN and control population.
How about these markers expression in normal urothelium near UCC in BEN? It is better to use normal urothelium of BEN as control?
Response: It is a great idea. We have a valuable collection of these tumors and we will do such a comparative analysis in future research.
- In patients and method: Does it justify to define patients from endemic settlements, the villages along the South Morava River basin as BEN tumors? Residents of rural and city areas treat as free of BEN?
Response: BEN diagnosis is extremely rare. Our tumors are from settlements that have been declared endemic according to all criteria for endemic diseases.
- In histology analysis: How to define necrosis?
Response: You will see in the revised manuscript that we clarified that.
The presence or absence of tumor necrosis was evaluated based on macroscopic and microscopic examination of tumor, and tumors were considered necrotic only if they exhibited more than 10% macroscopic necrosis, confirmed microscopically (14).
- The term TCC is not used currently instead of UCC.
Response: We have corrected that.
- The classification and grouping are really difficult to understand and be rational.
Response: We've simplified the groups.
- Tables are also poorly manifest. Table one and two with different format. Table 2 dif.st. not explain in footnote.
Response: Tables are simplified and formatted.
- Table 3&4 must be improved in format.
Response: Tables are simplified and formatted.
Kind regards,
Ana Ristic Petrovic
Reviewer 2 Report
Comments and Suggestions for Authors
I read with great interest the paper Expression of basal compartment and superficial markers in upper tract urothelial carcinoma associated with Balkan endemic nephropathy, a worldwide disease. I would like to commend with the authors in embarking in this kind of analysis.
This manuscript investigates the influence of CK5/6 and CD44, and superficial marker of urothelium - CK20 to pathological characteristics of UTUC in BEN and control population; and to determine the predictive influence of phenotypic characteristics in UTUC to expression of these markers.
The paper is well written and results are clearly described.
Authors should check minor typos and grammar
Authors could also rely (on citations 1-4) on the following manuscript in the introduction, adding ethnicity as a potential factor:
doi: 10.3390/biomedicines11071943.
Author Response
Dear Reviewer,
Thank you very much for all the comments and suggestions. All the revisions made will contribute to a better quality of the manuscript.
The paper is well written and results are clearly described.
Response: Thank you for reviewing our manuscript and for the suggestion.
Authors should check minor typos and grammar.
Response: Minor mistakes and typos were corrected.
Authors could also rely (on citations 1-4) on the following manuscript in the introduction, adding ethnicity as a potential factor:
doi: 10.3390/biomedicines11071943.
Response: We have read this excellent paper and cited it as you suggested
Tufano, A.; Perdonà, S.; Viscuso, P.; Frisenda, M.; Canale, V.; Rossi, A.; Del Prete, P.; Passaro, F.; Calarco, A. The Impact of Ethnicity and Age on Distribution of Metastases in Patients with Upper Tract Urothelial Carcinoma: Analysis of SEER Data. Biomedicines 2023, 11, 1943.
Kind regards,
Ana Ristic Petrovic
Round 2
Reviewer 1 Report
Comments and Suggestions for Authors
accepted after revision